# TaSIL: Taylor Series Imitation Learning

**Daniel Pfrommer**[*]
Massachusetts Institute of Technology[†]
Cambridge, MA
dpfrom@mit.edu

**Thomas T.C.K. Zhang**[*]
University of Pennsylvania
Philadelphia, PA
ttz2@seas.upenn.edu

**Stephen Tu**
Robotics at Google
New York, NY
stephentu@google.com

**Nikolai Matni**
University of Pennsylvania
Philadelphia, PA
nmatni@seas.upenn.edu

## Abstract

We propose Taylor Series Imitation Learning (TaSIL), a simple augmentation to standard behavior cloning losses in the context of continuous control. TaSIL penalizes deviations in the higher-order Taylor series terms between the learned and expert policies. We show that experts satisfying a notion of *incremental input-to-state stability* are easy to learn, in the sense that a small TaSIL-augmented imitation loss over expert trajectories guarantees a small imitation loss over trajectories generated by the learned policy. We provide generalization bounds for TaSIL that scale as $\tilde{\mathcal{O}}(1/n)$ in the realizable setting, for $n$ the number of expert demonstrations. Finally, we demonstrate experimentally the relationship between the robustness of the expert policy and the order of Taylor expansion required in TaSIL, and compare standard Behavior Cloning, DART, and DAgger with TaSIL-loss-augmented variants. In all cases, we show significant improvement over baselines across a variety of MuJoCo tasks.

## 1 Introduction

Imitation learning (IL), wherein expert demonstrations are used to train a policy [1, 2], has been successfully applied to a wide range of tasks, including self-driving cars [3, 4], robotics [5], and video game playing [6]. While IL is typically more sample-efficient than reinforcement learning-based alternatives, it is also known to be sensitive to distribution shift: small errors in the learned policy can lead to compounding errors and ultimately, system failure [3, 6]. In order to mitigate the effects of distribution shift caused by policy error, two broad approaches have been taken by the community. On-policy approaches such as DAgger [6] augment the data-set with expert-labeled or corrected trajectories generated by learned policies. In contrast, off-policy approaches such as DART [7] and GAIL [8] augment the data-set by perturbing the expert controlled system. In both cases, the goal is to provide examples to the learned policy of how the expert recovers from errors. While effective, these methods either require an interactive expert (on-policy), or access to a simulator for policy rollouts during training (off-policy), which may not always be practically feasible.

In this work, we take a more direct approach towards mitigating distribution shift. Rather than providing examples of how the expert policy recovers from error, we seek to endow a learned policy with the robustness properties of the expert directly. In particular, we make explicit the underlying assumption in previous work that a good expert is able to recover from perturbations through

---

[*]Both authors contributed equally to this work.
[†]This work was done while affiliated with the University of Pennsylvania.

36th Conference on Neural Information Processing Systems (NeurIPS 2022).

the notion of *incremental input-to-state stability*, a well-studied control theoretic notion of robust nonlinear stability. Under this assumption, we show that if the $p$-th order Taylor series approximation of the learned and expert policies approximately match on expert-generated trajectories, then the learned policy induces closed-loop behavior similar to that of the expert closed-loop system. Here the order $p$ is determined by the robustness properties of the expert, and makes quantitative the informal observation that more robust experts are easier to learn. Fundamentally, we seek to characterize settings where offline imitation learning is Probably Approximately Correctly (PAC)-learnable [9]; that is, defining notions of expert data and designing algorithms such that low training time error on offline expert demonstrations implies low test time error along the learned policy's trajectories, in spite of distribution shift.

**Contributions**  We propose and analyze Taylor Series Imitation Learning (TaSIL), which augments the standard behavior cloning imitation loss to capture errors between the higher-order terms of the Taylor series expansion of the learned and expert policies. We reduce the analysis of TaSIL to the analysis of a supervised learning problem over expert data: in particular, we identify a robustness criterion for the expert such that a small TaSIL-augmented imitation loss over expert trajectories guarantees that the difference between trajectories generated by the learned and expert policies is also small. We also provide a finite-difference based approximation that is applicable to experts that cannot directly query their higher-order derivatives, expanding the practical applicability of TaSIL. We show in the realizable setting that our algorithm achieves generalization bounds scaling as $\tilde{\mathcal{O}}(1/n)$, for $n$ the number of expert demonstrations. Finally, we empirically demonstrate (i) that the relationship between the robustness of the expert policy and the order of Taylor expansion required in TaSIL predicted by our theory is observed in practice, and (ii) the benefits of using the TaSIL-augmented imitation loss by comparing the sample-efficiency of standard and TaSIL-loss-augmented behavior cloning, DART, and DAgger on a variety of MuJoCo tasks: on hard instances where behavior cloning fails, the TaSIL-augmented variants show significant performance gains.

## 1.1 Related work

**Imitation learning**  Behavior cloning is known to be sensitive to compounding errors induced by small mismatches between the learned and expert policies [3, 6]. On-policy [6] and off-policy [7, 8] approaches exist that seek to prevent this distribution shift by augmenting the data-set created by the expert. While DAgger is known to enjoy $\tilde{\mathcal{O}}(T)$ sample-complexity in the task horizon $T$ for loss functions that are strongly convex in the policy parameters,[3] we are not aware of finite-data guarantees for DART or GAIL. In addition to these seminal papers, there is a body of work that seeks to leverage control theoretic techniques [10, 11] to ensure (robust) stability of the learned policy. More closely related to our work are the results by Ren et al. [12] and Tu et al. [13]. In Ren et al. [12], a two-stage pipeline of imitation learning followed by policy optimization via a PAC-Bayes regularization term is used to provide generalization bounds of the learned policy across randomly drawn environments. This work is mostly complementary to ours, as TaSIL could in principle be used to augment the imitation losses used in the first stage of their pipeline (leaving the second stage unmodified).

In Tu et al. [13], sample-complexity bounds for IL are provided under the assumption that the learned policy can be explicitly constrained to match the incremental stability properties of the expert. While conceptually appealing, practically enforcing such stability constraints is difficult, and as such Tu et al. [13] resort to heuristics in their implementation. In contrast, we provide sample-complexity guarantees for a practically implementable algorithm, and under much milder stability assumptions.

**Robust stability and learning for continuous control**  There is a rich body of work applying Lyapunov stability or contraction theory [14] to learning for continuous control. For example [15–18] use stability-based regularizers to trim the hypothesis space, and empirically show that this leads to more sample-efficient and robust learning methods. Lyapunov stability and contraction theory have also been used to provide finite sample-complexity guarantees for adaptive nonlinear control [19] and learning stability certificates [20].

---

[3]This bound degrades to $\tilde{\mathcal{O}}(T^2)$ when loss function is only convex, and does not hold for the nonconvex loss functions we consider.

## 2 Problem formulation

We consider the nonlinear discrete-time dynamical system

$$x_{t+1} = f(x_t, u_t), \ x_0 = \xi, \tag{1}$$

where $x_t \in \mathbb{R}^d$ is the system state, $u_t \in \mathbb{R}^m$ is the control input, and $f : \mathbb{R}^d \times \mathbb{R}^m \to \mathbb{R}^d$ defines the system dynamics. We study system (1) evolving under the control input $u_t = \pi(x_t) + \Delta_t$, for $\pi : \mathbb{R}^d \to \mathbb{R}^m$ a suitable control policy, and $\Delta_t \in \mathbb{R}^m$ an additive input perturbation (that will be used to capture policy errors). We define the corresponding *perturbed* closed-loop dynamics by $x_{t+1} = f_{\text{cl}}^\pi(x, \Delta_t) := f(x, \pi(x) + \Delta_t)$, and use $x_t^\pi(\xi, \{\Delta_s\}_{s=0}^{t-1})$ to denote the value of the state $x_t$ at time $t$ evolving under control input $u_t = \pi(x) + \Delta_t$ starting from initial condition $x_0 = \xi$. To lighten notation, we use $x_t^\pi(\xi)$ to denote $x_t^\pi(\xi, \{0\}_{s=0}^{t-1})$, and overload $\|\cdot\|$ to denote the Euclidean norm for vectors and the operator norm for matrices and tensors.

Initial conditions $\xi$ are assumed to be sampled randomly from a distribution $\mathcal{D}$ with support restricted to a compact set $\mathcal{X}$. We assume access to $n$ rollouts of length $T$ from an expert $\pi_\star$, generated by drawing initial conditions $\{\xi_i\}_{i=1}^n$ i.i.d. from $\mathcal{D}$. The IL task is to learn a policy $\hat{\pi}$ which leads to a closed-loop system with similar behavior to that induced by the expert policy $\pi_\star$ as measured by the *expected imitation gap* $\mathbb{E}_\xi \max_{1 \le t \le T} \left\| x_t^{\hat{\pi}}(\xi) - x_t^{\pi_\star}(\xi) \right\|$.

The baseline approach to IL, typically referred to as *behavior cloning* (BC), casts the problem as an instance of supervised learning. Denote the discrepancy between an evaluation policy $\bar{\pi}$ and the expert policy $\pi_\star$ on a trajectory generated by a rollout policy $\pi_d$ starting at initial condition $\xi$ by $\Delta_t^{\pi_d}(\xi; \bar{\pi}) := \bar{\pi}(x_t^{\pi_d}(\xi)) - \pi_\star(x_t^{\pi_d}(\xi))$. BC directly solves the supervised empirical risk minimization (ERM) problem

$$\hat{\pi}_{\text{bc}} \in \underset{\pi \in \Pi}{\text{argmin}} \ \frac{1}{n} \sum_{i=1}^n h(\{\Delta_t^{\pi_\star}(\xi_i; \pi)\}_{t=0}^{T-1}),$$

over a suitable policy class $\Pi$. Here, $h : (\mathbb{R}^m)^T \to \mathbb{R}$ is a loss function which encourages the discrepancy terms $\Delta_t^{\pi_\star}(\xi_i; \pi)$ to be small along the expert trajectories. While behavior cloning is conceptually simple, it can perform poorly in practice due to distribution shifts triggered by errors in the learned policy $\hat{\pi}_{\text{bc}}$. Specifically, due to the effects of compounding errors, the closed-loop system induced by the behavior cloning policy $\hat{\pi}_{\text{bc}}$ may lead to a dramatically different distribution over system trajectories than that induced by the expert policy $\pi_\star$, even when the population risk on the expert data, $\mathbb{E}_\xi h(\{\Delta_t^{\pi_\star}(\xi; \hat{\pi}_{\text{bc}})\}_{t=0}^{T-1})$, is small.

As described in the introduction, existing approaches to mitigating distribution shift seek to augment the data with examples of the expert recovering from errors. These approaches either require an interactive oracle (e.g., DAgger) or access to a simulator for policy rollouts (e.g., DART, GAIL), and may not always be practically applicable. To address the distribution shift challenge without resorting to data-augmentation, we propose TaSIL, an off-policy IL algorithm which provably leads to learned policies that are robust to distribution shift.

The rest of the paper is organized as follows: in Section 3, we focus on ensuring robustness to policy errors for a single initial condition $\xi$. We show that the imitation gap $\|x_t^\pi(\xi) - x_t^{\pi_\star}(\xi)\|$ between a test policy $\pi$ and the expert policy $\pi_\star$ can be controlled by the TaSIL-augmented imitation loss evaluated on the expert trajectory $\{x_t^{\pi_\star}(\xi)\}$, effectively reducing the analysis of the imitation gap to a supervised learning problem over the expert data. In Section 4, we integrate these results with tools from statistical learning theory to show that $n \gtrsim \varepsilon^{-r}/\delta$ trajectories are sufficient to achieve imitation gap of at most $\varepsilon$ with probability at least $1 - \delta$; here $r > 0$ is a constant determined by the stability properties of the expert policy $\pi_\star$, with more robust experts corresponding to smaller values of $r$. Finally, in Section 5, we validate our analysis empirically, and show that using the TaSIL-augmented loss function in IL algorithms leads to significant gains in performance and sample efficiency.

## 3 Bounding the imitation gap on a single trajectory

In this section, we fix an initial condition $\xi$ and test policy $\pi$, and seek to control the imitation gap $\Gamma_T(\xi; \pi) := \max_{1 \le t \le T} \|x_t^\pi(\xi, \{0\}) - x_t^{\pi_\star}(\xi, \{0\})\|$. A natural way to compare the

closed-loop behavior of a test policy $\pi$ to that of the expert policy $\pi_\star$ is to view the discrepancy $\Delta_t^\pi(\xi; \pi) := \pi(x_t^\pi(\xi)) - \pi_\star(x_t^\pi(\xi))$ as an *input perturbation* to the expert closed-loop system. By writing $x_{t+1} = f_{\mathsf{cl}}^\pi(x_t, 0) = f_{\mathsf{cl}}^{\pi_\star}(x_t, \Delta_t^\pi(\xi; \pi))$, the imitation gap can be written as $\Gamma_T(\xi; \pi) = \max_t \|x_t^{\pi_\star}(\xi, \{\Delta_s^\pi(\xi; \pi)\}_{s=0}^{t-1}) - x_t^{\pi_\star}(\xi, \{0\})\|$, suggesting that closed-loop expert systems that are robust to input perturbations, as measured by the difference between nominal and perturbed trajectories, will lead to learned policies that enjoy smaller imitation gaps.

Stability conditions defined in terms of differences between nominal and perturbed trajectories have been extensively studied in robust nonlinear control theory via the notion of *incremental input-to-state stability* ($\delta$-ISS) (see e.g., Angeli [21] and references therein). Before proceeding, we recall definitions of standard comparison functions [22]: a function $\gamma(x)$ is class $\mathcal{K}$ if it is continuous, strictly increasing, and satisfies $\gamma(0) = 0$, and a function $\beta(x, t)$ is class $\mathcal{KL}$ if it is continuous, $\beta(\cdot, t)$ is class $\mathcal{K}$ for each fixed $t$, and $\beta(x, \cdot)$ is decreasing for each fixed $x$.

**Definition 3.1** ($\delta$-ISS system). *Consider the closed-loop system evolving under policy $\pi$ and subject to perturbations $\Delta_t$ given by $x_{t+1} = f_{\mathsf{cl}}^\pi(x_t, \Delta_t)$. The closed-loop system $f_{\mathsf{cl}}^\pi(x_t, \Delta_t)$ is incremental input-to-state stable ($\delta$-ISS) if there exists a class $\mathcal{KL}$ function $\beta$ and a class $\mathcal{K}$ function $\gamma$ such that for all initial conditions $\xi_1, \xi_2 \in \mathcal{X}$, perturbation sequences $\{\Delta_t\}_{t\geq 0}$, and $t \in \mathbb{N}$:*

$$\left\| x_t^\pi(\xi_1; \{\Delta_s\}_{s=0}^{t-1}) - x_t^\pi(\xi_2; \{0\}_{s=0}^{t-1}) \right\| \leq \beta(\|\xi_1 - \xi_2\|, t) + \gamma\left( \max_{0 \leq k \leq t-1} \|\Delta_k\| \right). \qquad (2)$$

Definition 3.1 says that: (i) trajectories generated by $\delta$-ISS systems converge towards each other if they begin from different initial conditions, and (ii) the effect of bounded perturbations $\{\Delta_t\}$ on trajectories is bounded. Our results will only require the stability conditions of Definition 3.1 to hold for a class of norm bounded perturbations. In light of this, we say that a system is $\eta$-locally $\delta$-ISS if equation (2) holds for all input perturbations satisfying $\sup_{t\in\mathbb{N}} \|\Delta_t\| \leq \eta$.

By writing $x_{t+1} = f_{\mathsf{cl}}^\pi(x_t, 0) = f_{\mathsf{cl}}^{\pi_\star}(x_t, \Delta_t^\pi(\xi; \pi))$, we conclude that if $x_{t+1} = f_{\mathsf{cl}}^{\pi_\star}(x_t, \Delta_t)$ is $\eta$-locally $\delta$-ISS , and that $\sup_{t\in\mathbb{N}} \|\Delta_t^\pi(\xi; \hat{\pi})\| \leq \eta$, then equation (2) yields

$$\|x_t^\pi(\xi) - x_t^{\pi_\star}(\xi)\| \leq \gamma\left( \max_{0 \leq k \leq t-1} \|\Delta_k^\pi(\xi; \pi)\| \right) \implies \Gamma_T(\xi; \pi) \leq \gamma\left( \max_{0 \leq t \leq T-1} \|\Delta_t^\pi(\xi; \pi)\| \right). \quad (3)$$

Equation (3) shows that the imitation gap $\|x_t^\pi(\xi) - x_t^{\pi_\star}(\xi)\|$ is controlled by the maximum discrepancy $\|\Delta_t^\pi(\xi; \pi)\|$ incurred on trajectories generated by the policy $\pi$. A natural way of bounding the test discrepancy $\|\Delta_t^\pi(\xi; \pi)\|$, defined over trajectories generated by $\pi$, by the training discrepancy $\|\Delta_t^{\pi_\star}(\xi; \pi)\|$, defined over trajectories generated by $\pi_\star$, is to write out a Taylor series expansion of the former around the latter, i.e., to write:[4]

$$\|\Delta_t^\pi(\xi; \pi)\| \leq \|\Delta_t^{\pi_\star}(\xi; \pi)\| + \|\partial_x \Delta_t^{\pi_\star}(\xi; \pi)\| \Gamma_T(\xi; \pi) + \mathcal{O}\left( \Gamma_T^2(\xi, \pi) \right), \qquad (4)$$

where $\partial_x \Delta_t^{\pi_\star}(\xi; \pi)$ denotes the partial derivative of the discrepancy with respect to the argument of the policies $\pi$ and $\pi_\star$. The challenge however, is that the imitation gap $\Gamma_T(\xi; \pi)$ also appears in the Taylor series expansion (4), leading to an implicit constraint. We show that this can be overcome if the order $p$ of the Taylor series expansion (4) is sufficiently large, as determined by the decay rate of the class $\mathcal{K}$ function $\gamma(\cdot)$ defining the robustness of the expert policy. We begin by identifying a condition reminiscent of adversarially robust training objectives that ensures a small imitation gap.

**Proposition 3.1.** *Let the expert closed-loop system $f_{\mathsf{cl}}^{\pi_\star}$ be $\eta$-locally $\delta$-ISS for some $\eta > 0$. Fix an imitation gap bound $\varepsilon > 0$, initial condition $\xi$, and policy $\pi$. Then if*

$$\max_{0 \leq t \leq T-1} \sup_{\|\delta\| \leq \varepsilon} \|\pi_\star(x_t^{\pi_\star}(\xi) + \delta) - \pi(x_t^{\pi_\star}(\xi) + \delta)\| \leq \min\{\eta, \gamma^{-1}(\varepsilon)\}, \qquad (5)$$

*we have that the imitation gap satisfies $\Gamma_T(\xi; \pi) \leq \varepsilon$.*

Proposition 3.1 states that if a policy $\pi$ is sufficiently close to the expert policy $\pi_\star$ in a tube around the expert trajectory, then the imitation gap remains small. How to ensure that inequality (5) holds using only offline data from expert trajectories is not immediately obvious. The Taylor series expansion (4) suggests that a natural approach to satisfying this condition is to match derivatives of the test policy $\pi$ to those of the expert policy $\pi_\star$. We show next that a sufficient order for such a Taylor series

---

[4]We only take a first order expansion here for illustrative purposes.

expansion is naturally determined by the decay rate of the class $\mathcal{K}$ function $\gamma(x)$ towards 0. Less robust experts have functions that decay to 0 more slowly and will lead to stricter sufficient conditions. Conversely, more robust experts have functions that decay to 0 more quickly, and will lead to more relaxed sufficient conditions. We focus on two disjoint classes of class $\mathcal{K}$ functions: (i) functions that decay in their argument faster than a linear function, i.e. $\gamma(x) < \mathcal{O}(x)$ as $x \to 0^+$, and (ii) functions that decay in their argument no faster than a linear function, i.e. $\gamma(x) \geq \Omega(x)$ as $x \to 0^+$.

**Rapidly decaying class $\mathcal{K}$ functions**

We show that when the class $\mathcal{K}$ function $\gamma(x)$ decays to 0 faster than $\mathcal{O}(x)$ in some neighborhood of 0, then matching the *zeroth-order difference* $\max_t \|\Delta_t^{\pi_\star}(\xi; \pi)\|$ on the expert trajectory, as is done in vanilla behavior cloning, suffices to close the imitation gap. We make the following assumption on the test policy $\pi$ and expert policy $\pi_\star$.

**Assumption 3.1.** *There exists a non-negative constant $L_\pi$ such that $\|\bar{\pi}(x) - \bar{\pi}(y)\|_2 \leq L_\pi \|x - y\|_2$ for all $x$, $y$, and $\bar{\pi} \in \{\pi, \pi_\star\}$.*

Proposition 3.1 then leads to the following guarantee on the imitation gap.

**Theorem 3.1.** *Fix a test policy $\pi$ and initial condition $\xi \in \mathcal{X}$, and let Assumption 3.1 hold. Let $f_{\mathsf{cl}}^{\pi_\star}$ be $\eta$-locally $\delta$-ISS for some $\eta > 0$, and assume that the class $\mathcal{K}$ function $\gamma(\cdot)$ in (2) satisfies $\gamma(x) \leq \mathcal{O}(x^{1+r})$ for some $r > 0$. Choose constants $\mu, \alpha > 0$ such that*

$$2L_\pi x + (x/\mu)^{\frac{1}{1+r}} \leq \gamma^{-1}(x) \text{ for all } 0 \leq x \leq \alpha. \tag{6}$$

*Provided that the imitation error on the expert trajectory incurred by $\pi$ satisfies:*

$$\max_{0 \leq t \leq T-1} \mu \|\Delta_t^{\pi_\star}(\xi; \pi)\|^{1+r} \leq \alpha, \quad \max_{0 \leq t \leq T-1} 2L_\pi \mu \|\Delta_t^{\pi_\star}(\xi; \pi)\|^{1+r} + \|\Delta_t^{\pi_\star}(\xi; \pi)\| \leq \eta, \tag{7}$$

*then for all $1 \leq t \leq T$ the instantaneous imitation gap is bounded as*

$$\|x_t^{\pi_\star}(\xi) - x_t^\pi(\xi)\| \leq \max_{0 \leq k \leq t-1} \mu \|\Delta_k^{\pi_\star}(\xi; \pi)\|^{1+r}. \tag{8}$$

Theorem 3.1 shows that if a policy $\pi$ is a sufficiently good approximation of the expert policy $\pi_\star$ on an expert trajectory $\{x_t^{\pi_\star}(\xi)\}$, then the imitation gap $\|x_t^{\pi_\star}(\xi) - x_t^\pi(\xi)\|$ can be upper bounded in terms of the discrepancy term $\max_{0 \leq k \leq t-1} \|\Delta_t^{\pi_\star}(\xi; \pi)\|$ *evaluated on the expert trajectory*. To help illustrate the effect of the decay parameter $r$ on the choices of $\mu$ and $\alpha$, we make condition (6) more explicit by assuming that $\gamma(x) \leq Cx^{1+r}$ for all $x \in [0, 1]$. Then one can choose $\mu = 2^{1+r}C$ and $\alpha = \mathcal{O}(1)(L_\pi^{1+r}C)^{-1/r}$. This makes clear that a larger $r$, i.e., a more robust expert, leads to less restrictive conditions (7) on the policy $\pi$ and a tighter upper bound on the imitation gap (8) (assuming $\max_{0 \leq k \leq t-1} \|\Delta_t^{\pi_\star}(\xi; \pi)\| < 1$). In particular, for such systems, vanilla behavior cloning is sufficient to ensure bounded imitation gap. This also makes clear how the result breaks down when $r \approx 0$, i.e., when the decay rate is nearly linear, as the neighborhood $\alpha$ can become arbitrarily small, such that it may be impossible to learn a policy $\pi$ that satisfies the bound (7) with a practical number of samples $n$. The interplay of the bounds (7) in Theorem 3.1 (and the subsequent theorems of its like) and the sample-complexity of imitation learning will be discussed in further detail in Section 4.

**Slowly decaying class $\mathcal{K}$ functions**

When the class $\mathcal{K}$ function $\gamma(x)$ decays to 0 slowly, for reasons discussed above, controlling the zeroth-order difference $\Delta_t^{\pi_\star}(\xi; \pi)$ may not be sufficient to bound the imitation gap. In particular, we consider class $\mathcal{K}$ functions satisfying $\gamma(x) \leq \mathcal{O}(x^{1/r})$ for some $r \geq 1$. Setting $p = \lfloor r \rfloor$, we now show that matching up to the $p$-th total derivative of $\pi_\star$ is sufficient to control the imitation gap. Analogously to Assumption 3.1, we make the following regularity assumption on the test policy $\pi$ and expert policy $\pi_\star$.

**Assumption 3.2.** *For a given non-negative $p \in \mathbb{N}$, assume that the test policy $\pi$ and expert policy $\pi_\star$ are $p$-times continuously differentiable, and there exists a constant $L_{\partial^p \pi} \geq 0$ such that*

$$\left\| \bar{\pi}(x) - \left( J_{x_0}^p \bar{\pi} \right)(x) \right\| \leq \frac{L_{\partial^p \pi}}{(p+1)!} \|x - x_0\|^{p+1}, \tag{9}$$

*for all $x, x_0$ and $\bar{\pi} \in \{\pi, \pi_\star\}$, where $\left( J_{x_0}^p \bar{\pi} \right)(x) := \sum_{j=0}^p \frac{1}{j!} \left( \partial_x^j \bar{\pi}(x_0) \right)(x - x_0)^{\otimes j}$ is the $p$-th order Taylor polynomial of $\bar{\pi}$ evaluated at $x_0$, and $\otimes$ denotes the tensor product.*

With this assumption in hand, we provide the following guarantee on the imitation gap.

**Theorem 3.2.** *Let $f_{\mathsf{cl}}^{\pi_\star}$ be $\eta$-locally $\delta$-ISS for some $\eta > 0$, and assume that the class $\mathcal{K}$ function $\gamma(\cdot)$ in (2) satisfies $\gamma(x) \leq \mathcal{O}(x^{1/r})$ for some $r \geq 1$. Fix a test policy $\pi$ and initial condition $\xi \in \mathcal{X}$, and let Assumption 3.2 hold for $p \in \mathbb{N}$ satisfying $p + 1 - r > 0$. Choose $\mu, \alpha > 0$ such that*

$$2\frac{L_{\partial^p \pi}}{(p+1)!}x^{p+1} + (x/\mu)^r \leq \gamma^{-1}(x), \text{ for all } 0 \leq x \leq \alpha \leq \frac{1}{2}. \tag{10}$$

*Provided the jth total derivatives, $j = 0, \ldots, p$, of the imitation error on the expert trajectory incurred by $\pi$ satisfy:*

$$\max_{0 \leq t \leq T-1} \max_{0 \leq j \leq p} \mu\left(\frac{2}{j!}\left\|\partial_x^j \Delta_t^{\pi_\star}(\xi; \pi)\right\|\right)^{1/r} \leq \alpha, \tag{11}$$

$$\max_{0 \leq t \leq T-1} \max_{0 \leq j \leq p} \frac{2L_{\partial^p \pi}\mu^{p+1}}{(p+1)!}\left(\frac{2}{j!}\left\|\partial_x^j \Delta_t^{\pi_\star}(\xi; \pi)\right\|\right)^{\frac{p+1}{r}} + \frac{2}{j!}\left\|\partial_x^j \Delta_t^{\pi_\star}(\xi; \pi)\right\| \leq \eta, \tag{12}$$

*then for all $1 \leq t \leq T$ the instantaneous imitation gap is bounded by*

$$\left\|x_t^{\pi_\star}(\xi) - x_t^\pi(\xi)\right\| \leq \max_{0 \leq k \leq t-1} \max_{0 \leq j \leq p} \mu\left(\frac{2}{j!}\left\|\partial_x^j \Delta_t^{\pi_\star}(\xi; \pi)\right\|\right)^{1/r}. \tag{13}$$

Theorem 3.2 shows that if the $p$-th order Taylor series of the policy $\pi$ approximately matches that of the expert policy $\pi_\star$ when evaluated on an expert trajectory $\{x_t^{\pi_\star}(\xi)\}$, then the imitation gap $\left\|x_t^{\pi_\star}(\xi) - x_t^\pi(\xi)\right\|$ can be upper bounded in terms of the derivatives of the discrepancy term, i.e., by $\max_{0 \leq k \leq t-1} \max_{0 \leq j \leq p} \left\|\partial_x^j \Delta_k^{\pi_\star}(\xi; \pi)\right\|$, *evaluated on the expert trajectory*. To help illustrate the effect of the choice of the order $p$ on the constants $\mu$ and $\alpha$, we make condition (10) more explicit by assuming that $\gamma(x) \leq Cx^{1/r}$ for all $x \in [0, 1]$. Then one can choose $\mu = 2^{1/r}C$ and $\alpha = \mathcal{O}(1)(L_{\partial^p \pi}C^r)^{-1/(p+1-r)}$. This expression highlights a tradeoff: by picking larger order $p$, the right hand side $\alpha$ of bound (11) increases, but at the expense of having to match higher-order derivatives. This also highlights that both the order $p$ and closeness required by Equation (11) get increasingly restrictive as $r$ increases, matching our intuition that less robust experts lead to harder imitation learning problems.

**Using estimated derivatives**   We show in Appendix C that the results of Theorems 3.1 and 3.2 extend gracefully to when only approximate derivatives $\widehat{\partial_x^j \pi_\star}(x)$ can be obtained, e.g., through finite-difference methods. In particular, if $\|\widehat{\partial_x^j \pi_\star}(x) - \partial_x^j \pi_\star(x)\| \leq \varepsilon$, then it suffices to appropriately tighten the constraints (11) and (12) by $\mathcal{O}(\varepsilon^{1/r})$ and $\mathcal{O}(\varepsilon)$, respectively. Please refer to Appendix C for more details.

## 4   Algorithms and generalization bounds for TaSIL

The analysis of Section 3 focused on a single test policy $\pi$ and initial condition $\xi$. Theorems 3.1 and 3.2 motivate defining the $p$-TaSIL loss function:

$$\ell_p^{\pi_\star}(\xi; \pi) := \frac{1}{p+1}\sum_{j=0}^p \max_{0 \leq t \leq T-1} \left\|\partial_x^j \Delta_t^{\pi_\star}(\xi; \pi)\right\|. \tag{14}$$

The corresponding policy $\hat{\pi}_{\mathsf{TaSIL},p}$ is the solution to the empirical risk minimization (ERM) problem:

$$\hat{\pi}_{\mathsf{TaSIL},p} \in \operatorname*{argmin}_{\pi \in \Pi} \frac{1}{n}\sum_{i=1}^n \ell_p^{\pi_\star}(\xi_i; \pi), \tag{15}$$

which explicitly seeks to learn a policy $\pi \in \Pi$, for $\Pi$ a suitable policy class, that matches the $p$-th order Taylor series expansion of the expert policy.[5] In this section, we analyze the generalization and sample-complexity properties of the $p$-TaSIL ERM problem (15).

---

[5]Although we focus on the supremum loss $\max_{0 \leq t \leq T-1}\left\|\partial_x^j \Delta_t^{\pi_\star}(\xi; \pi)\right\|$ in our analysis, we note that any surrogate loss that upper bounds the supremum loss, e.g., $\sum_{t=0}^{T-1}\left\|\partial_x^j \Delta_t^{\pi_\star}(\xi; \pi)\right\|$, can be used.

Our analysis in this section focuses on the *realizable* setting: we assume that for every dataset of expert trajectories $\{\{x_t^{\pi_\star}(\xi_i)\}_{t=0}^{T-1}\}_{i=1}^n$, there exists a policy $\pi \in \Pi$ that achieves (near) zero empirical risk. This is true if, for example, $\pi_\star \in \Pi$. In this setting, we demonstrate that we can attain generalization bounds that decay as $\tilde{\mathcal{O}}(n^{-1})$, where $\tilde{\mathcal{O}}(\cdot)$ hides poly-log dependencies on $n$. These rates are referred to as *fast rates* in statistical learning, since they decay faster than the $n^{-1/2}$ rate prescribed by the central limit theorem. We present analysis for the non-realizable setting in Appendix B: this analysis is standard and yields generalization bounds scaling as $\mathcal{O}(n^{-1/2})$.

Let $\mathcal{G} \subset [0,1]^{\mathcal{X}}$ be a set of functions mapping some domain $\mathcal{X}$ to $[0,1]$.[6] Let $\mathcal{D}$ be a distribution with support restricted to $\mathcal{X}$, and denote the mean of $g \in \mathcal{G}$ with respect to $x \sim \mathcal{D}$ by $\mathbb{E}_x[g]$. Similarly, fixing data points $x_1, \ldots, x_n \in \mathcal{X}$, we denote the empirical mean of $g$ by $\mathbb{E}_n[g] := n^{-1} \sum_{i=1}^n g(x_i)$. We focus our analysis on the following class of parametric Lipschitz function classes.

**Definition 4.1** (Lipschitz parametric function class)**.** *A parametric function class $\mathcal{G} \subset [0,1]^{\mathcal{X}}$ is called $(B_\theta, L_\theta, q)$-Lipschitz if $\mathcal{G} = \{g_\theta(\cdot) \mid \theta \in \Theta\}$ with $\Theta \subset \mathbb{R}^q$, and it satisfies the following boundedness and uniform Lipschitz conditions:*

$$\sup_{\theta \in \Theta} \|\theta\| \leq B_\theta, \quad \sup_{x \in \mathcal{X}} \sup_{\theta_1, \theta_2 \in \Theta, \theta_1 \neq \theta_2} \frac{|g_{\theta_1}(x) - g_{\theta_2}(x)|}{\|\theta_1 - \theta_2\|} \leq L_\theta. \tag{16}$$

*We assume without loss of generality that $B_\theta L_\theta \geq 1$.*

The description (16) is very general, and as we show next, is compatible with feed-forward neural networks with differentiable activation functions. We then have the following generalization bound, which adapts [23, Corollary 3.7] to Lipschitz parametric function classes using the machinery of local Rademacher complexities [23]. Alternatively, the result can also be derived from [24, Theorem 3].

**Theorem 4.1.** *Let $\mathcal{G} \subset [0,1]^{\mathcal{X}}$ be a $(B_\theta, L_\theta, q)$-Lipschitz parametric function class. There exists a universal positive constant $K < 10^6$ such that the following holds. Given $\delta \in (0,1)$, with probability at least $1 - \delta$ over the i.i.d. draws $x_1, \ldots, x_n \sim \mathcal{D}$, for all $g \in \mathcal{G}$, the following bound holds:*

$$\mathbb{E}_x[g] \leq 2\mathbb{E}_n[g] + K \left( \frac{q \log(B_\theta L_\theta n) + \log(1/\delta)}{n} \right). \tag{17}$$

We now use Theorem 4.1 to analyze the generalization properties of the $p$-TaSIL ERM problem (15). In what follows, we assume that the expert-closed loop system is stable in the sense of Lyapunov, i.e., that there exists $B_X > 0$ such that $\sup_{t \in \mathbb{N}, \xi \in \mathcal{X}} \|x_t^{\pi_\star}(\xi)\| \leq B_X$, and consider the following parametric class of $p + 2$ continuously differentiable policies:

$$\Pi_{\theta,p} := \{\pi(x, \theta) \mid \theta \in \mathbb{R}^q, \|\theta\| \leq B_\theta, \pi(0, \theta) = 0 \,\forall \theta, \pi \text{ is } p+2 \text{ continuously differentiable}\}. \tag{18}$$

Define the constants

$$B_j := \sup_{\|x\| \leq B_X, \|\theta\| \leq B_\theta} \left\| \partial_x^j \pi(x, \theta) \right\|, \quad L_j := \sup_{\|x\| \leq B_X, \|\theta\| \leq B_\theta} \left\| \partial_x^{j+1} \partial_\theta \pi(x, \theta) \right\|,$$

for $j = 0, \ldots, p$, and note that they are guaranteed to be finite under our regularity assumptions. Finally, define the loss function class:

$$\ell_p^{\pi_\star} \circ \Pi_{\theta,p} := \left\{ \ell_p^{\pi_\star}(\cdot; \pi) \text{ defined in } (14) \mid \pi \in \Pi_{\theta,p} \right\}. \tag{19}$$

From a repeated application of Taylor's theorem, we show in Lemma B.2 that $B_{\ell,p}^{-1}(\ell_p^{\pi_\star} \circ \Pi_{\theta,p})$ is a $(B_\theta, B_{\ell,p}^{-1} L_{\ell,p}, q)$-Lipschitz parametric function class for $B_{\ell,p} := \frac{2}{p+1} \sum_{j=0}^p B_j$ and $L_{\ell,p} := \frac{B_X}{p+1} \sum_{j=0}^p L_j$. We now combine this with Theorem 4.1 to bound the population risk achieved by the solution to the TaSIL ERM problem (15).

**Corollary 4.1.** *Let the policy class $\Pi_{\theta,p}$ be defined as in (18), and assume that $\pi_\star \in \Pi_{\theta,p}$. Let the function class $\ell_p^{\pi_\star} \circ \Pi_{\theta,p}$ be defined as in (19), and constants $B_{\ell,p}, L_{\ell,p}$ be defined as above. Let $\hat{\pi}_{\mathsf{TaSIL},p}$ be any empirical risk minimizer (15). Then with probability at least $1 - \delta$ over the initial conditions $\{\xi_i\}_{i=1}^n \overset{\text{i.i.d.}}{\sim} \mathcal{D}^n$,*

$$\mathbb{E}_\xi\left[\ell_p^{\pi_\star}(\xi; \hat{\pi}_{\mathsf{TaSIL},p})\right] \leq \mathcal{O}(1) B_{\ell,p} \frac{q \log\left(B_\theta B_{\ell,p}^{-1} L_{\ell,p} n\right) + \log(1/\delta)}{n}. \tag{20}$$

---

[6]This is without loss of generality for $[0,B]$-bounded functions by considering the normalized function class $B^{-1}\mathcal{G} := \{B^{-1}g \mid g \in \mathcal{G}\} \subset [0,1]^{\mathcal{X}}$.

We note that since these generalization bounds solely concern the supervised learning problem of matching the expert on the expert trajectories, the constants do not depend on the stability properties of the trajectories generated by the learned policy $\hat{\pi}_{\mathsf{TaSIL},p}$. To convert the generalization bound (20) to a probabilistic bound on the imitation gap $\Gamma_T(\xi; \hat{\pi}_{\mathsf{TaSIL},p})$, we first apply Markov's inequality to bound the probability that the conditions of Theorem 3.1 or 3.2 hold by the expected TaSIL loss (20), and then apply Corollary 4.1 together with Markov's inequality.

**Theorem 4.2** (Rapidly decaying class $\mathcal{K}$ functions). *Assume that $\pi_\star \in \Pi_{\theta,0}$ and let the assumptions of Theorem 3.1 hold for all $\pi \in \Pi_{\theta,0}$. Let Equation (6) hold with constants $\mu, \alpha > 0$, and assume without loss of generality that $\alpha/\mu \leq 1$, $L_\pi \mu \geq 1/2$. Let $\hat{\pi}_{\mathsf{TaSIL},0}$ be an empirical risk minimizer of $\ell_0^{\pi_\star}$ over the policy class $\Pi_{\theta,0}$ for initial conditions $\{\xi_i\} \overset{\text{i.i.d.}}{\sim} \mathcal{D}^n$. Fix a failure probability $\delta \in (0,1)$, and assume that*

$$n \geq \mathcal{O}(1) \max\left\{ B_{\ell,0} \frac{\kappa_\alpha}{\delta} \log\left( \frac{\kappa_\alpha B_\theta L_{\ell,0}}{\delta} \right), \ B_{\ell,0} \frac{\kappa_\eta}{\delta} \log\left( \frac{\kappa_\eta B_\theta L_{\ell,0}}{\delta} \right) \right\},$$

*where $\kappa_\alpha := q(\mu/\alpha)^{1/(1+r)}$, $\kappa_\eta := qL_\pi\mu/\eta$. Then with probability at least $1 - \delta$, the imitation gap evaluated on $\xi \sim \mathcal{D}$ (drawn independently from $\{\xi_i\}_{i=1}^n$) satisfies*

$$\Gamma_T(\xi; \hat{\pi}_{\mathsf{TaSIL},0}) \leq \mathcal{O}(1) \, \mu \left( \frac{1}{\delta} \frac{B_{\ell,0} q \log\left( B_\theta B_{\ell,0}^{-1} L_{\ell,0} n \right)}{n} \right)^{1+r}.$$

**Theorem 4.3** (Slowly decaying class $\mathcal{K}$ functions). *Assume that $\pi_\star \in \Pi_{\theta,p}$, and let the assumptions of Theorem 3.2 hold for all $\pi \in \Pi_{\theta,p}$. Let Equation (10) hold with constants $\mu, \alpha > 0$. Let $\hat{\pi}_{\mathsf{TaSIL},p}$ be an empirical risk minimizer of $\ell_p^{\pi_\star}$ over the policy class $\Pi_{\theta,p}$ for initial conditions $\{\xi_i\} \overset{\text{i.i.d.}}{\sim} \mathcal{D}^n$. Fix a failure probability $\delta \in (0,1)$, and assume*

$$n \geq \mathcal{O}(1) \max_{j \leq p} \max\left\{ B_j \frac{\kappa_{\alpha,j}}{\delta} \log\left( \frac{\kappa_{\alpha,j} B_\theta B_j^{-1} B_X L_j}{\delta} \right), \ B_j \frac{\kappa_{\eta,j}}{\delta} \log\left( \frac{\kappa_{\eta,j} B_\theta B_j^{-1} B_X L_j}{\delta} \right) \right\},$$

*where $\kappa_{\alpha,j} := \left( \frac{\mu}{\alpha} \right)^r \frac{pq}{j!}$ and $\kappa_{\eta,j} := \left( \frac{L_{\partial^p \pi}}{(p+1)!} \frac{\mu^{p+1}}{(j!)^{(p+1)/r}} + \frac{1}{j!} \right) \frac{pq}{\eta\delta}$. Then with probability at least $1 - \delta$, the imitation gap evaluated on $\xi \sim \mathcal{D}$ (drawn independently from $\{\xi_i\}_{i=1}^n$) satisfies*

$$\Gamma_T(\xi; \hat{\pi}_{\mathsf{TaSIL},p}) \leq \mathcal{O}(1) \, \mu \max_{j \leq p} \left( \frac{p}{j!\delta} \frac{B_j q \log\left( B_\theta B_j^{-1} B_X L_j n \right)}{n} \right)^{1/r}.$$

In the rapidly decaying setting, corresponding to more robust experts, Theorem 4.2 states that $n \gtrsim \varepsilon^{-\frac{1}{1+r}}/\delta$ expert trajectories are sufficient to ensure that the imitation gap $\Gamma_T(\xi; \hat{\pi}_{\mathsf{TaSIL},0}) \lesssim \varepsilon$ with probability at least $1 - \delta$. Recall that more robust experts have larger values of $r > 0$, leading to smaller sample-complexity bounds. In contrast, to achieve the same guarantees on the imitation gap in the slowly decaying setting, Theorem 4.3 states $n \gtrsim \varepsilon^{-r}/\delta$ expert trajectories are sufficient, where we recall that less robust experts have larger values of $r \geq 1$. These theorems quantitatively show how the robustness of an underlying expert affects the sample-complexity of IL, with more robust experts enjoying better dependence on $\varepsilon$ than less robust experts. We note that analogous dependencies on expert stability are reflected in the burn-in requirements, i.e., the number of expert trajectories required to ensure with high probability no catastrophic distribution shift occurs, of each theorem.

## 5 Experiments

We compare three standard imitation learning algorithms, Behavior Cloning, DAgger, and DART, to TaSIL-augmented loss versions. In TaSIL-augmented algorithms, we replace the standard imitation loss function

$$\ell_{\mathsf{IL}}^{\pi_\star}(\{\xi_i\}_{i=1}^n; \pi) := \tfrac{1}{n} \sum_{i=1}^n \sum_{t=0}^{T-1} \|\Delta_t^{\pi_\star}(\xi_i; \pi)\|$$

with the $p$-TaSIL-augmented loss

$$\ell_{\mathsf{TaSIL},p}^{\pi_\star}(\{\xi_i\}_{i=1}^n; \pi) := \tfrac{1}{n} \sum_{i=1}^n \sum_{t=1}^{T-1} \sum_{j=0}^p \lambda_j \|\mathrm{vec}(\partial_x^j \Delta_t^{\pi_\star}(\xi_i; \pi))\|,$$

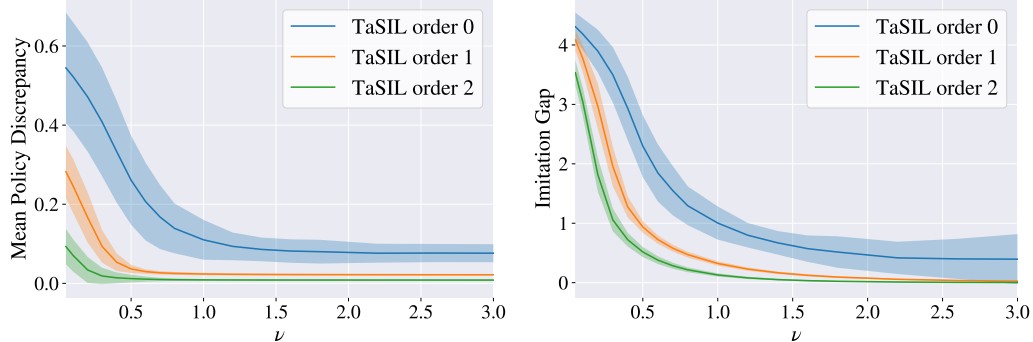

**Figure 1: Left:** The average Euclidean norm difference between the expert and learned policies on trajectories rolled out under the learned policy. **Right:** The maximum deviation between an expert and learned policy trajectory starting from identical initial conditions. All statistics are averaged across 50 test trajectories and we plot the mean and standard deviation for 10 random seeds.

where $\{\lambda_j\}_{j=0}^p$ are positive tunable regularization parameters. We use the Euclidean norm of the vectorized error in the derivative tensors as more optimizer-amenable surrogate to the operator norm.

We experimentally demonstrate (i) the effect of the expert stability properties and order of the TaSIL loss, and (ii) that TaSIL-loss-based imitation learning algorithms are more sample efficient than their standard counterparts. All experiments[7] are carried out using Jax [25] GPU acceleration and automatic differentiation capabilities to compute the higher-order derivatives, and the Flax [26] neural network and Optax [27] optimization toolkits.

**Stability Experiments**   To illustrate the effect of the expert closed-loop system stability on sample-complexity, we consider a simple $\delta$-ISS stable dynamical system with a tunable $\gamma$ input-to-state gain function. For state and input $x_t, u_t \in \mathbb{R}^{10}$, the dynamics are:

$$x_{t+1} = \eta x_t + (1 - \eta) \frac{\gamma(\|h(x_t) + u_t\|)}{\|h(x_t) + u_t\|}(h(x_t) + u_t).$$

The perturbation function $h : \mathbb{R}^{10} \to \mathbb{R}^{10}$ is set to a randomly initialized MLP with two hidden layers of width 32 and GELU [28] activations such that the expert $\pi_\star(x) = -h(x)$ yield a closed loop system $f_{cl}^{\pi_\star}(x, \Delta)$ which is $\delta$-ISS stable with the specified class $\mathcal{K}$ function $\gamma$ (see Appendix D). We use $\eta = 0.95$ for all experiments presented here.

We investigate the performance of $p$-TaSIL loss functions for $\delta$-ISS system with different class $\mathcal{K}$ stability. We sweep $\mathcal{K}$ functions $\gamma(x) = Cx^\nu$ for $\nu \in [0.05, 3]$, $C = 5$ and $p$-TaSIL loss functions for $p \in \{0, 1, 2\}$ (additional details can be found in Appendix D). The results of this sweep are shown in Figure 1. Higher-order $p$-TaSIL losses significantly reduce both the imitation gap and the mean policy discrepancy on test trajectories. Notably, the first and second order TaSIL loss maintain their improved performance for slower decaying class $\mathcal{K}$ functions. Theorem 3.1 and Theorem 3.2 yield lower bounds of $\nu = 1$, $\nu = 2^{-1}$, and $\nu = 3^{-1}$ for closing the imitation gap using the 0-TaSIL, 1-TaSIL, and 2-TaSIL losses respectively. Figure 1 demonstrates significant performance degradation in policy discrepancy and decaying imitation gap starting around these threshold values.

**MuJoCo Experiments**   We evaluate the ability of the TaSIL loss to improve performance on standard imitation learning tasks by modifying Behavior Cloning, DAgger [6], and DART [7] to use the $\ell_{\mathsf{TaSIL},1}$ loss and testing them in simulation on different OpenAI Gym MuJoCo tasks [29]. The MuJoCo environments we use and their corresponding (state, input) dimensions are: Walker2d-v3 (17, 6), HalfCheetah-v3 (17, 6), Humanoid-v3 (376, 17), and Ant-v3 (111, 8).

For all environments we use pretrained expert policies obtained using Soft Actor Critic reinforcement learning by the Stable-Baselines3 [30] project. The experts consist of Multi-Layer Perceptrons with two hidden layers of 256 units each and ReLU activations. For all environments, learned policies have 2 hidden layers with 512 units each and GELU activations in addition to Batch Normalization. The final policy output for both the expert and learned policy are rescaled to the valid action space

---

[7]The code used for these experiments can be found at https://github.com/unstable-zeros/TaSIL

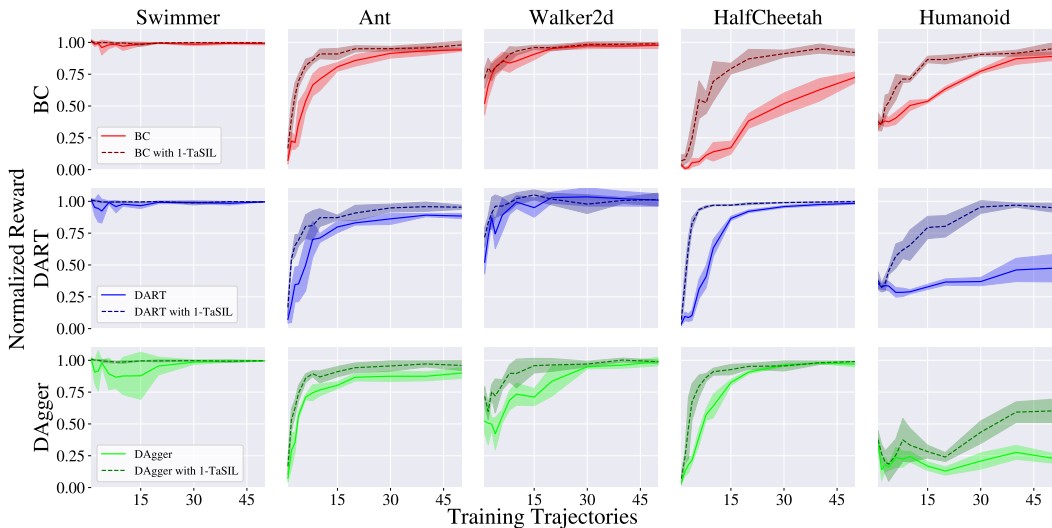

**Figure 2:** Cumulative expert-normalized rewards as a function of trajectory budget for policies trained using different algorithms with and without 1-TaSIL loss.

after applying a tanh nonlinearity. We used trajectories of length $T = 300$ for all experiments. We refer to Appendix E for additional experiment details.

In Figure 2 we report the mean expert-normalized rewards across 5 seeds for all algorithms and environments as a function of the trajectory budget provided. Algorithms with 1-TaSIL loss showed significant improvement in sample-complexity across all challenging environments. The expert for the Swimmer environment is very robust due to the simplicity of the task, and so as predicted by Theorem 3.1 and 4.2, all algorithms (with the exception of the vanilla DAgger algorithm due to it initially selecting poor rollout trajectories) are able to achieve near expert performance across all trajectory budgets. We are also able to nearly match or exceed the performance of standard on-policy methods DAgger and DART with our off-policy 1-TaSIL loss Behavior Cloning in all environments. Additional experimental results can be found in the appendix. These include representative videos of the behavior achieved by the expert, BC, and TaSIL-augmented BC policies in the supplementary material, where once again, a striking improvement is observed, especially in low-data regimes for harder environments, as well as a systematic study of finite-difference-based approximations of 1-TaSIL which achieve comparable performance to Jacobian-based implementations.

## 6   Conclusion

We presented Taylor Series Imitation Learning (TaSIL), a simple augmentation to behavior cloning that penalizes deviations in the higher-order Taylor series terms between the learned and expert policies. We showed that $\delta$-ISS experts are easier to learn, both in terms of the loss-function that needs to be optimized and sample-complexity guarantees. Finally, we showed the benefit of using TaSIL-augmented losses in BC, DAgger, and DART across a variety of MuJoCo tasks. This work opens up many exciting future directions, including extending TaSIL to pixel-based IL, and to (offline/inverse) reinforcement learning settings.

## Acknowledgements

We thank Vikas Sindhwani, Sumeet Singh, Jean-Jacques E. Slotine, Jake Varley, and Fengjun Yang for helpful feedback. Nikolai Matni is supported by NSF awards CPS-2038873, CAREER award ECCS-2045834, and a Google Research Scholar award.

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
